# Tissue–Resident Memory T Cells in Chronic Inflammation—Local Cells with Systemic Effects?

**DOI:** 10.3390/cells10020409

**Published:** 2021-02-16

**Authors:** Anoushka Ashok Kumar Samat, Jolijn van der Geest, Sebastiaan J. Vastert, Jorg van Loosdregt, Femke van Wijk

**Affiliations:** 1Center for Translational Immunology, University Medical Center Utrecht, Utrecht University, 3584 CX Utrecht, The Netherlands; A.A.K.Samat-3@umcutrecht.nl (A.A.K.S.); j.i.e.vandergeest@students.uu.nl (J.v.d.G.); b.vastert@umcutrecht.nl (S.J.V.); j.vanloosdregt@umcutrecht.nl (J.v.L.); 2Paediatric Rheumatology and Immunology, Wilhelmina Children’s Hospital, Utrecht University, 3584 EA Utrecht, The Netherlands

**Keywords:** tissue–resident memory T cells, re(circulation), chronic inflammation

## Abstract

Chronic inflammatory diseases such as rheumatoid arthritis (RA), Juvenile Idiopathic Arthritis (JIA), psoriasis, and inflammatory bowel disease (IBD) are characterized by systemic as well as local tissue inflammation, often with a relapsing-remitting course. Tissue–resident memory T cells (T_RM_) enter non-lymphoid tissue (NLT) as part of the anamnestic immune response, especially in barrier tissues, and have been proposed to fuel chronic inflammation. T_RM_ display a distinct gene expression profile, including upregulation of CD69 and downregulation of CD62L, CCR7, and S1PR1. However, not all T_RM_ are consistent with this profile, and it is now more evident that the T_RM_ compartment comprises a heterogeneous population, with differences in their function and activation state. Interestingly, the paradigm of T_RM_ remaining resident in NLT has also been challenged. T cells with T_RM_ characteristics were identified in both lymph and circulation in murine and human studies, displaying similarities with circulating memory T cells. This suggests that re-activated T_RM_ are capable of retrograde migration from NLT via differential gene expression, mediating tissue egress and circulation. Circulating ‘ex-T_RM_’ retain a propensity for return to NLT, especially to their tissue of origin. Additionally, memory T cells with T_RM_ characteristics have been identified in blood from patients with chronic inflammatory disease, leading to the hypothesis that T_RM_ egress from inflamed tissue as well. The presence of T_RM_ in both tissue and circulation has important implications for the development of novel therapies targeting chronic inflammation, and circulating ‘ex-T_RM_’ may provide a vital diagnostic tool in the form of biomarkers. This review elaborates on the recent developments in the field of T_RM_ in the context of chronic inflammatory diseases.

## 1. Introduction

### 1.1. T_RM_: Drivers of Chronic Inflammation

Characteristic of chronic inflammatory diseases is their self-perpetuating nature. Chronic inflammation can affect virtually any organ or tissue in the body including skin, gut, joint, muscles, and the central nervous system. Tissue–resident memory T cells (T_RM_) are known to be canonically present in tissues, playing a vital role in immune responses, but are also suspected of contributing adversely to chronic inflammatory conditions [1]. Specifically, T_RM_ have been implicated in the hyper-response in inflamed environments due to their potential capability of cross-talk with other immune cells. Murine studies have demonstrated the ‘immune sentinel’ function of T_RM_ by examining the functional response of these cells in mouse models of vitiligo and colitis, respectively [1,2]. The production of immune cell recruiting cytokines and expression of genes involved in the recruitment of innate immune cells supports the hypotheses that T_RM_ have a crucial role in chronic inflammation by adding to the immune-inflammatory state [1,2]. This review aims to summarize the recent developments in the field of T_RM_ with a focus on their role in chronic inflammation. Additionally, the concept of T_RM_ recirculation and the impact this may have on exacerbation of chronic inflammatory diseases will be discussed.

### 1.2. T_RM_ Phenotype

T_RM_ are typically present in every tissue compartment and can be found residing in even higher numbers in barrier tissues such as the skin and gut [3]. The key markers of T_RM_ are considered to be the C-type lectin CD69, and the αE integrin CD103 [1]. However not all T_RM_ populations express CD103, and it is mainly expressed by CD8^+^ T_RM_ and to a lesser extent CD4^+^ T_RM_. Possibly, CD103 expression is enriched in certain non-lymphoid tissue (NLT) compartments, such as the epithelium [1,4]. CD69 antagonizes S1PR1, which is required for migration of T cells out of tissues, hence S1PR1 inhibition facilitates tissue-residency [4]. In murine tissues, T_RM_ populations that do not express CD69 have also been identified [4]. The variation of CD69 and CD103 expression on T_RM_ between tissues indicates that T_RM_ might comprise of subpopulations that differ between tissues.

Although transcriptional and functional T_RM_ core signatures have been identified [3,5] there is considerable heterogeneity of T_RM_ populations reflected in surface markers, cytokine/chemokine, and transcription factor expression. Therefore, it is important to use multiple markers to identify T_RM_ and to include their functional profile. Furthermore, heterogeneity has not only been observed between T_RM_ populations from different tissues, but also within T_RM_ population in one tissue compartment. For example, a comparative study found that the majority of T_RM_ in human skin were CD4^+^CD69^+^CD103^–^ cells (dermis), with a mixed epidermal population (CD4^+^CD103^+^ and CD8^+^CD103^+^). Skin from naïve mice showed CD103^+/−^CD4^+^ T_RM_ to be most prevalent, with a large proportion of dermal CD4^+^ memory T cells retaining their ability to circulate [5].

Studies in secondary lymphoid tissues (SLTs) of both humans and mice demonstrate that CD4^+^ and CD8^+^ T_RM_ are present in lymph nodes (LNs) and the spleen, with up to 30–50% of these cells expressing phenotypes and transcriptional profiles compatible with T_RM_ cells [3,6,7]. In the human LN, CD8^+^ T_RM_ display phenotypic, functional, and epigenetic signatures associated with tissue residency, while exhibiting a higher proliferative capacity as well. However, T_RM_ also exhibit an organ-specific signature compared with other sites, with increased expression of TCF-1, LEF-1, CXCR5, and CXCR4, and reduced expression of effector molecules [8]. This suggests that the LN niche is a site for extended T_RM_ maintenance and quiescence.

These studies identify distinct ‘subsets’ of T_RM_, some displaying more effector capacity and others displaying more proliferative capacity Taken together, T_RM_ are a highly heterogeneous pool of cells and presumably capable of plasticity as well as homing, allowing them to exist in a range of phenotypes. Other surface markers and transcription factors that are part of the characteristic T_RM_ expression profile are summarized in Table 1.

### 1.3. T_RM_ Development

A study from 2016 showed that in early life, memory T cells (mostly T_EM_) were found in higher frequencies in mucosal barrier tissues than in secondary lymphoid organs (SLOs) and circulation, compared to young adults where T_EM_ cells were predominant in all mucosal sites (>90%). The majority of T_EM_ cells in all pediatric tissue are CD69^+^, while circulating T_EM_ were CD69^−^. A lower frequency of pediatric mucosal CD8^+^ T_EM_ co-express CD69 and CD103 (compared to >90% in adult mucosa), implying that early life memory T cells in mucosa are not fully differentiated T_RM_ cells. As such, the sequestering of early life memory T cells to mucosal barrier tissue and not to draining LN is suggestive of ‘local in-situ priming’ to inhaled and ingested antigens [15]. The inkling is that inhalation and ingestion of food and the microbiome itself may results in antigen triggers that prime memory T cells in early life and lead to T_RM_ seeding of tissue compartments. There is also a consideration of tissue repletion. Once tissue compartments have been filled (early life seeding of T_RM_), a competitive dynamic is established, and strong triggers (infections) are required to direct further seeding. Memory precursor T cells and/or T_EM_ cells both may play a role in the seeding dynamics. Thus, inflammation may not be the only driving force behind seeding of tissue compartments especially in early life, though it is one of the established mechanisms, especially in terms of T_EM_ cells.

Memory T cell subsets consist of circulating and non-circulating subsets, the former of which comprises T_CM_ and T_EM_, recirculating through lymphoid and non-lymphoid tissues respectively [16]. In the event of pathogen challenge, the non-circulating T_RM_ in peripheral tissues are poised for an immune response [17]. Recently, T_RM_ have been shown to potentially repopulate the lymphoid T_CM_ and T_EM_ compartments, alluding to the possibility that the development of these cells may be underscored by an epigenetic program, allowing developmental plasticity and an imprinted memory of prior localization [18,19].

T_RM_ development is presumed to be driven by antigen exposure, as well as cytokines and chemokines in the local tissue environment. TGFβ has been shown to drive T_RM_ generation (by downregulation of transcription factors Eomes and T-bet), as well as induce CD103 expression [20]. Recent studies suggest that T_RM_ can sustain themselves in the tissue. For example, it was shown that T_RM_ proliferated in situ upon re-stimulation with antigen in the female reproductive tract as well as the skin of mice, respectively [21,22]. IL-15 is thought to be essential for long-term survival of T_RM_ [11]. This retention and survival of T_RM_ dependent on the local availability of cytokines and factors (creating organ-specific dependencies) may influence the ability of egress of these cells during an immune challenge [11].

T-bet dependent IL-15 signaling has been shown to induce expression of the transcription factor Hobit in mice [12]. The dependence on these cytokines varies between tissues, as reviewed by an extensive comparative study [23]. The transcription factors Hobit, Blimp1, and Runx3 were shown to regulate differentiation of CD8^+^ T_RM_ in mice, by stimulating expression of genes required for tissue residency and suppressing genes that mediate tissue egress [12,13]. A comparison between the gene signature of mouse T_RM_ with human T_RM_ showed that most genes followed a similar expression pattern [3]. However, there are also clear discrepancies between T_RM_ in mice and humans. Hobit, for instance, is a key transcription factor that drives T_RM_ function in mice [12,24] but expression was observed to be variable in human T_RM_, and may not be differentially expressed in T_RM_ compared with other memory T cells [3,25,26]. Vieira Braga et al. observed increased Hobit expression in human effector-like human CD8^+^ T cells, but not in naive or memory CD8^+^ T cells [27]. This raises the possibility that T_RM_ with an effector phenotype express increased Hobit, but other T_RM_ subsets do not.

To truly differentiate between activated (CD69^+^CD62L^−^) effector T cells and CD69^+^ T_RM_ cells, it may be crucial to perform full transcriptome analysis at a single cell level, as well as T cell receptor sequencing and analysis of other phenotypic markers. In inflamed tissues an expression gradient of CD69 (CD69hi vs. CD69lo) may discern T_RM_ from T_EM_, alongside other markers specific to the two different pools of cells, but this has to be proven. The key cytokines and transcription factors that drive T_RM_ differentiation are summarized in Figure 1. More recent studies have identified roles for other transcription factors and epigenetic changes in the differentiation of CD8^+^ T_RM_, which are reviewed by Chen et al. [28].

## 2. T_RM_ Are Enriched at Sites of Chronic Inflammation at Barrier and Non-Barrier Tissues

In humans, various CD4^+^ and CD8^+^ T_RM_ (sub)populations were found to be enriched in inflammatory diseases that manifest at barrier tissues, such as inflammatory bowel disease (IBD) in the gut [24] and psoriasis in the skin [29], but also in disease manifested in non-barrier tissues, including diabetes type I [30], Sjogren’s syndrome [31], lupus nephritis [32], multiple sclerosis (MS) [33,34] and several arthritic diseases [35,36,37,38,39] such as Juvenile Idiopathic Arthritis (JIA) [35,36], psoriatic arthritis [37], ankylosing spondylitis [38], and rheumatoid arthritis (RA) [39]. The recent findings on T_RM_ in inflammatory diseases in various tissues is summarized in Table 2 below.

### 2.1. Skin

For chronic inflammatory diseases of the skin, such as vitiligo and psoriasis, a role for T_RM_ in re-occurring lesions and disease flares has long been suspected [29,49]. The markers that are most widely used to identify T_RM_ in the skin are CD69, CD103, and CD49a. A study looking into lesional and non-lesional skin samples from psoriasis patients found that the epidermis with active psoriasis was massively infiltrated by CD8^+^ T_RM_, both compared to non-lesional skin and healthy skin, with a 100-fold increase of T_RM_ in active psoriatic lesions [40]. Moreover, significant enrichment of CD103^+^CD8^+^ T_RM_ in lesional psoriatic skin compared to healthy controls was also observed [41]. Similarly, identification of T_RM_ populations in vitiligo patients portrayed that CD8^+^ T_RM_ were present in both dermis and epidermis of vitiligo skin and CD69^+^CD103^+^CD8^+^ T_RM_ were observed to be significantly enriched in vitiligo patients (independent from disease status) compared to psoriatic skin and healthy controls [42]. Furthermore, a significant increase of melanocyte-reactive T_RM_ in the lesional skin of vitiligo patients was noted [43].

Recent studies have also investigated T_RM_ populations in other autoimmune skin disorders. A novel study set up to identify T_RM_ in the skin of atopic dermatitis (AD) patients used single cell RNA sequencing to analyse both lesional and non-lesional skin, as well as skin from healthy controls. CD69^+^CD103^+^CD8^+^ T cells, but not CD69^+^CD103^+^CD4^+^ T cells, were significantly expanded in lesional skin from AD patients in comparison with non-lesional AD skin and healthy controls [44]. Additionally, in alopecia areata (AA), a disease caused by attack of the bulb region of the hair follicle by autoreactive CD8+ T cells, higher numbers of peribulbar CD103^+^ CD8^+^ T cells and increased CD69^+^ and CD103^+^ T cell numbers were found to be present in lesional skin from AA patients compared to non-lesional skin and healthy controls [45,46,50].

Consistent with the hypothesis of a role for T_RM_ in the mediation of chronic inflammation, increased production of pro-inflammatory cytokines was found in T_RM_ populations at lesional sites. A study observed that CD103^+^CD8^+^ T cells in psoriatic skin expressed IL-17 and IL-22 mRNA [40], noting that particularly CD49a^−^CD103^+^CD8^+^ T_RM_ in the skin predominantly produced IL-17 upon stimulation, and that this subset was enriched in psoriasis [51]. Another study had similar findings, with T cells in explants of lesional psoriatic skin that were activated with pan-T cell activating antibody OKT-3 poised for IL-17A production [52]. It was also demonstrated that CD49a^+^CD103^+^CD8^+^ T_RM_ in the skin of vitiligo patients had increased capability of IFN-γ production. Moreover, these cells recognized melanocyte-derived antigens. Increased expression of granzyme B and perforin was also found. Ex vivo, more co-expression of granzyme B and perforin was observed in CD49a^+^CD103^+^CD8^+^ T_RM_ compared to CD49a^−^ T_RM_, suggesting higher cytotoxic capacity in CD49a^+^ T_RM_ [51]. An examination of T_EM_ from perilesional skin of vitiligo patients found that these T_EM_ included T_RM_ populations. The T_RM_ from skin of vitiligo patients with active disease displayed increased IFN-γ and TNF-α production compared to stable disease and healthy controls, suggesting that T_RM_ in vitiligo are poised for production of these cytokines [42].

Taking the example of psoriasis, skin with clinically resolved psoriasis was demonstrated to contain increased IL-17 mRNA expressing CD103^+^CD8^+^ T_RM_, and an enrichment of IL-23 responsive CD103^+^CCR6^+^IL23R^+^CD8^+^ T cells [40]. In addition, pathogenic IL17A-producing T cell clones were found to be present in both active psoriatic plaques as well as in resolved skin lesions, but not in healthy control skin [53]. Moreover, in disease-naïve non-lesional skin of psoriatic patients, an enrichment of IL-17A and IFN-γ producing CCR6^+^ T cells [54], and IL-17A producing CD103^+^CD8^+^ T cells [41], compared to healthy controls was observed. Together these studies demonstrate that IL-17 and IFN-γ producing resident T cells are present in non-affected skin as well as resolved psoriatic lesions, and that these T_RM_ cells are poised for re-initiation of inflammation. Therefore, T_RM_ may play a role in the initiation of psoriasis pathogenesis as well as disease flares [41,54].

### 2.2. Gut

CD69^+^CD103^+^ T_RM_ are also numerous in the intestine, with mostly CD8^+^ T_RM_ in the intestinal epithelium and both CD4^+^ T_RM_ and CD8^+^ T_RM_ in the lamina propria [4]. Inflammatory bowel disease (IBD) is a chronic inflammatory disease of the intestine, and includes Crohn’s Disease (CD) and ulcerative colitis (UC). IBD is thought to be caused by an overactive immune response against the gut microbiome, and is characterized by relapsing flares, which suggests that T_RM_ may be involved in the pathogenesis [24].

Two studies extensively investigated the presence of T_RM_ in IBD. Bishu et al. investigated colon tissue (both epithelium and lamina propria) from CD patients, discerning CD4^+^ T_RM_ using CD69^+^ and CCR7^−^ expression. CD4^+^ T_RM_ constituted the majority of mucosal memory CD4^+^ T cells and the CD4^+^ T_RM_ compartment was expanded, with absolute numbers increased in CD patients compared to controls. CD8^+^ T_RM_ cells were not found to be enriched in CD patients compared to controls [53]. Colon samples from patients with active CD showed an increase in IL-17A producing CD4^+^ T_RM_, portraying a T_H_17 signature. CD4^+^ T_RM_ were also found to be the major source of TNF-α production in CD patients ex vivo [47]. Zundler et al. investigated the presence of T_RM_ in IBD patients using core characteristics of T_RM_ (as summarized in Table 1) and observed enrichment of CD69^+^CD4^+^ T cells in the lamina propria of IBD patients with increased expression of CXCR6, CD103 and CD101, and decreased expression of KLF2 [24]. They also demonstrated that gut CD69^+^ T cells (from IBD patients) produce elevated amounts of the pro-inflammatory cytokines IFN-γ, IL-13, IL-17A, and TNF-α compared to CD69^−^ T cells. In addition, a high number of CD4^+^ intestinal T_RM_ was associated with shorter flare-free survival in these patients [24]. In line with this it was noted that gut-resident CD4^+^ T cells produced increased IL-17 compared to circulating CD4^+^ T cells, which was even more evident in patients with IBD [55]. CD161^+^CD4^+^ T cells in CD patients were found to display elevated production of IL-17 and IL-22, and higher expression of IL-23R, as is characteristic of a T_H_17 phenotype (stimulated by IL-1b and IL-23) [56]. Taken together, these studies show that CD4^+^ T_RM_ in IBD patients have increased capabilities for the production of pro-inflammatory cytokines, including IL-17A and TNF-α, and may drive disease flares of IBD.

The role of CD8^+^ T_RM_ in IBD has been studied less extensively. A study on CD8^+^ T_RM_ in healthy gut samples demonstrated that the CD103^+^CD8^+^ T_RM_ in the lamina propria expressed granzyme B and perforin and were potent producers of IFN-γ, IL-2, and TNF-α upon activation. This suggests that CD8^+^ T_RM_ participate in the protective immune response of the gut [57]. CD8^+^ T_RM_ numbers were also found to be significantly reduced in the intestinal epithelium of IBD patients with quiescent disease compared to healthy controls, postulating that the decreased barrier immunity in IBD pathogenesis may be due to T_RM_ deficiency [58]. This touches upon the dual role T_RM_ can have in inflammatory disease: on one hand they play a role in tissue homeostasis and integrity, protecting the host from overzealous inflammation, on the other hand they may sustain inflammation once a perpetual loop of inflammation has been instigated.

### 2.3. Joints

Arthritic inflammatory diseases are characterized by chronic, recurring inflammation in the joints. In JIA, a form of chronic arthritis that begins in children before the age of sixteen, there is accumulation of both CD4^+^ and CD8^+^ T cells in the synovium of inflamed joints. Petrelli et al. examined the CD8^+^ T cell population in the synovial fluid (SF) of JIA patients, and observed that a PD-1 expressing subset of CD8^+^ T cells was highly enriched in comparison with peripheral blood (PB) of JIA patients and healthy controls [35]. The PD-1^+^CD8^+^ T cell subset displayed a hallmark T_RM_ transcriptional profile with upregulation of ITGAE, ITGA1, and CXCR6, and downregulation of S1PR1, KLF3, and SELL, compared to SF PD-1^−^CD8^+^ T cells. Gene set enrichment analysis (GSEA) showed that the PD-1^+^CD8^+^ T cells from JIA SF do not display an exhausted gene signature but instead are enriched for signature effector genes compared to their PD-1^−^ counterparts. PD-1^+^CD8^+^ T cells showed increased expression of Ki-67, and increased expression of pro-inflammatory cytokines IFN-γ, TNF-α, and granzyme B. In addition, PD-1^+^CD8^+^ T cells were clonally expanded, and their TCRs barely overlapped with PD-1^−^CD8^+^ T cells, suggesting that these are local antigen-driven pathogenic cells [35].

Recently, subsets of CD4^+^ T cells in the SF of JIA patients were also identified displaying a T_RM_ phenotype. Characteristic T_RM_ gene patterns included low CD62L and CCR7 expression and increased DUSP6, ITGAE (encoding CD103), CXCR6, and PD-1 expression [36]. The PD-1^+^CD4^+^ SF T_RM_ were found to secrete GM-CSF and IL-21, and these cells were shown to be enriched not only in the SF of JIA, but in the inflamed intestine of IBD patients as well. This suggests that PD-1^+^CD4^+^ T_RM_ participate in chronic inflammation in several diseases. In line with the data of Petrelli et al., PD-1^+^CD8^+^ T cells present in the same SF samples were also demonstrated to produce GM-CSF and IL-21. Therefore, both PD-1^+^CD8^+^ and PD1^+^CD4^+^ T_RM_ populations seem to have the potential to drive chronic inflammation in JIA [36], and joint or tissue inflammation in general. Steel et al. investigated the phenotype of IL-17A expressing CD8^+^ T cells, Tc17 cells, which were enriched in the SF of actively inflamed joints from patients with psoriatic arthritis compared to PB. Virtually all of the IL-17A^+^CD8^+^ T cells in the SF displayed a memory phenotype, and 65% expressed PD-1. Gene profiling showed that IL-17A^+^CD8^+^ T cells expressed genes characteristic of the T_RM_ phenotype compared to IL-17A^+^CD4^+^ T cells. These hallmarks include increased expression of ITGAE (encoding CD103) and ZNF683 (encoding the transcription factor Hobit), CXCR6 and decreased expression of S1PR1 [37]. Using flow cytometry, they further confirmed high CD103 expression on IL-17A^+^CD8^+^ T cells and these cells often co-expressed β7 integrin (a gut homing marker), cutaneous lymphocyte antigen (CLA) and CD49a (skin homing markers). Finally, examination of the relationship between CD69 and CD103 expression and IL-17A production demonstrated that CD69^−^CD103^−^CD8^+^ T cells contained minimal IL-17A producing T cells, whereas CD69 and/or CD103 expressing cells contained higher fractions of IL-17A producing T cells [37]. A large fraction of the IL-17A-producing cells also expressed pro-inflammatory cytokines IFN-γ and/or TNF-α and high levels of granzyme B.

Analysis of the SF of patients with ankylosing spondylitis identified a distinct population of memory CD103^+^β7^+^CD49a^+^CD29^+^ CD8^+^ T cells that was enriched in the SF of ankylosing spondylitis patients compared to PB. The β7 integrin was often co-expressed with CD103, and CD29 was often co-expressed with CD49a, thus allowing for identification of these cells using only CD103 and CD49a expression. Enrichment of CD49a^+^CD103^+^CD8^+^ T cells was not observed in the SF compared to PB in RA patients. CD49a^+^CD103^+^CD8^+^ T cells displayed a T_RM_-like transcriptional profile, with elevated expression of IL-10 and CXCR6, and a lack of CD62L and S1PR1 [38]. It was demonstrated that the T_RM_ produced IL-17A and granzyme B in both resting and stimulated conditions, and IL-10 and TNF-α when stimulated.

Very recently, a study identified a population of T cells that expressed T_RM_ characteristics using flow cytometry, enriched in the SF of adult RA patients. The cells in this population were memory CD69^+^CD103^+^ or CD103^−^CD8^+^ T cells. Protein expression of these cells was consistent with the characteristic T_RM_ phenotype, including increased CXCR6, CD49a, and CD101 expression, and decreased S1PR1 and KLF2 expression compared to CD69^−^CD8^+^ T cells. PD-1 and Ki-67 expression were also significantly elevated in these CD69^+^CD8^+^ T cells [39]. A significant increase of IFN-γ and TNF-α expression in T_RM_ upon stimulation with anti-CD3 antibody was observed, although this increase was seen in CD69^−^CD8^+^ T cells as well. However, T_RM_ displayed higher levels of granzyme B and perforin expression than CD69^−^CD8^+^ T cells, which further increased when cells were stimulated with IL-15, but not when stimulated with anti-CD3 antibody [39]. In summary, it has been demonstrated that SF from inflamed joints in arthritic inflammatory diseases contains highly activated CD4^+^ and CD8^+^ T_RM_ with pro-inflammatory properties. It should be noted that the SF is an exudate of the inflamed synovium, not truly a tissue. Identifying T_RM_ in SF is therefore not an entirely direct proof of the involvement of T_RM_ in the tissue inflammation. However, SF can be accessed much less invasively than synovial tissue [35,36,37,38,48,59] and SF T cells have been shown to display overlapping phenotypic and functional profiles with synovial tissue derived T_RM_ [35,36,37,60,61].

## 3. Recirculation of Tissue–Resident Memory T Cells

### 3.1. T_RM_ Migrate Out of Non-Lymphoid Tissue into Circulation

In the context of chronic inflammation, the occurrence of disease flares or relapses is evident in multiple diseases such as JIA (multiple joints affected in the same patient) and CD (multiple inflammatory patches). The tendency of recurrent inflammatory flares and expansion of the disease to, sometimes anatomically separated, locations, has prompted the hypothesis of potential T_RM_ recirculation. Are recirculating T_RM_ cells indeed capable of triggering recurrent inflammatory bursts as well as potentially spreading disease to ‘non-affected’ areas of the body? Most of our knowledge about T_RM_ recirculation comes from animal studies. Early indications for the ability of T_RM_ to egress from non-lymphoid tissue (NLT) came from the study of Beura et al., in which they investigated the origin and transcriptional profile of secondary lymphoid organ (SLO) T_RM_ [62]. They used a C57BL/6J mouse model, into which naive lymphocytic choriomeningitis virus (LCMV)-specific transgenic CD8^+^ T cells were transferred. Subsequently, these mice were infected with lymphocytic choriomeningitis virus (LCMV) of the Armstrong strain, inducing an acute infection. After clearance of the LCMV infection, the LCMV-specific P14 CD8^+^ memory T cells were examined. A CD69^+^CD62Llo tissue resident population of P14 CD8^+^ memory T cells was observed in all examined SLOs. Tissue residency of this CD8^+^ T cell population was confirmed with a parabiosis experiment.

Further observation of these T_RM_ in lymph nodes (LNs) showed that local re-stimulation of NLT resulted in accumulation of these cells in draining LN, but not in distant LN. These data indicate that NLT CD8^+^ T_RM_ migrate from NLT into LN upon reactivation, and are a source of SLO T_RM_. Administration of a molecule that inhibits S1PR1 resulted in a reduction of T_RM_ in draining LN after reactivation, suggesting a role for S1PR1 in tissue egress by T_RM_ [18]. Finally, it was demonstrated that T_RM_ could differentiate into T_CM_, by upregulation of CD62L. Downregulation of CD69 and CD103 was observed on these ‘ex-T_RM_’ as well. Of note, traces of the T_RM_ phenotype did remain. For instance, CCR9 expression was only slowly downregulated and still higher on ex-T_RM_ than in other circulating cells 100 days after infection [18]. Finally, in another study the same group also identified a population of CD4^+^ T_RM_ in SLO of Armstrong LMCV infected mice, which shared transcriptional characteristics with CD4^+^ T_RM_ from NLT [63].

Fonseca et al. further examined the egress of CD8^+^ T_RM_ from NLT in mice. They generated C57BL/6J mice with OT-I T_RM_ in their skin, i.e., ovalbumin specific CD8^+^ T cells. After reactivation of these T_RM_, skin-derived cells could be found in both draining and distant LN. Furthermore, circulating T cells, but not T_RM_, were depleted in this mouse model. Ten days after re-stimulation of T_RM_, OT-I T cells were observed in the blood. These cells displayed transient retention of a T_RM_ phenotype (CD103 expression and lack of CD62L expression), thus these cells were referred to as ex-T_RM_. Similar results were obtained when reactivating T_RM_ in the female reproductive tract instead of the skin [58]. Additionally, it was demonstrated that T_RM_ intravenously transferred into recipient mice were observed in the same tissue compartment in the recipient as in the donor. This suggests that T_RM_ have a predilection for returning to their parental tissue. Bias for return to the tissue compartment of origin was still observed in ex-T_RM_ in which a tertiary immune response was induced, suggesting maintenance of an epigenetic T_RM_ profile. This indicates that ex-T_RM_ have a propensity to re-acquire a T_RM_ phenotype [18].

Another study examined T_RM_ in human skin, defined as CD69^+^CD103^+^CLA^+^ T cells. Tissue explant cultures with human skin were performed, and a CD69- population of T cells were detected. Tissue exit was associated with downregulation of CD69 though less pronounced than in the CD8^+^ T cell compartment. Egress of CD103^+^ CLA^+^CD4^+^ T cells was also studied in vivo using human skin transplants on immune-deficient mice. After 50 days, the T cell population of the spleen was analyzed and the same population was observed [64]. This suggest that skin T_RM_ had egressed from the human skin explant in this mouse model as well. CD103^+^CLA^+^CD4^+^ T cells were abundant in the skin in both epidermis and dermis, but constituted less than 2% of CLA^+^CD4^+^ memory T cells in circulation. The very same population was also observed in lymph of the human thoracic duct, indicating a CD4^+^ T cell population in the circulation of humans that mirrors a population of skin T_RM_ [64]. Based on these data, it can be postulated that CD4^+^ T_RM_ egress from NLT via LN into circulation, similar to CD8^+^ T_RM_. Together, these studies indicate that tissue egress by T_RM_ occurs in both mice and humans, in the CD8^+^ as well as CD4^+^ T_RM_ compartment.

### 3.2. Ex-T_RM,_ Present in the Circulation, Share Characteristics with T_RM_ in Tissue, But Also Display Molecules Characteristic of Circulating T Cells

After identification of T_RM_-like cells in SLOs and circulation, one study used mass cytometry to analyze the phenotype of blood CLA^+^ ex-T_RM_. A specific cluster, comprised of CD103^+^CLA^+^CD4^+^ T cells, was observed within the CLA^+^CD4^+^ T cell population. Cells of this cluster expressed CCR4, CCR6 and CCR10, which are indicative for skin tropism, and CD101 and β7 integrin, while lacking CD27, CCR7 and CXCR3 expression [64]. This phenotype was shared by CD103^+^CLA^+^CD4^+^ T cells in the skin. Additional transcriptional profiling demonstrated that blood CD103^+^CLA^+^CD4^+^ T cells displayed a distinct transcriptional signature (compared to memory CD4^+^CLA^+^CD103^−^CCR^−^ and CD4^+^CLA^+^CD103^−^CCR7^+^ T cells in circulation), which was also significantly enriched in skin CD103^+^CLA^+^CD4^+^ T cells. The shared transcriptional profile confirmed the phenotype observed with mass cytometry, and further included increased expression of ITGAE, CD101, CXCR6 and TWIST1, and decreased expression of Eomes. Moreover, TCRβ sequencing of CLA^+^CD4^+^ memory T cells showed that CD103^+^CLA^+^CD4^+^ T cells from skin and blood are strongly clonally related, whereas little overlap was found with other CLA^+^CD4^+^ memory T cell populations. Finally, a lack of CD69 expression was observed in CD4^+^ ex-T_RM_ in the blood, while this molecule was highly expressed on CD4^+^ T_RM_ in the skin. This implicates that skin T_RM_ were able to egress from tissue by downregulating CD69 [64].

A murine study examined the phenotype of SLO CD8^+^ T_RM_ using the markers CD103, CD122, Ly6C, CD27, CD62L, CXCR3, CCR9, KLRG1, CX3CR1, CD69, and CD44. Based on these markers, SLO T_RM_ were shown to be more similar to NLT T_RM_ (from either skin, small intestine or female reproductive tract) than to T_CM_ and T_EM_, although SLO T_RM_ did comprise a population distinct from NLT T_RM_. SLO CD8^+^ T_RM_ shared part of the characteristic T_RM_ gene expression profile with NLT T_RM_, whereas expression of other genes was more closely related to T_CM_. Thus, SLO CD8^+^ T_RM_ were demonstrated to be a memory T cell population that is similar to but distinct from NLT T_RM_ and shares characteristics with T_CM_ [62].

These findings further support the notion that both human and mouse T_RM_ are capable of tissue egress and that ex-T_RM_ have undergone changes in gene expression that mediate egress and circulation, for example downregulation of CD69 and transient upregulation of CD62L. As such, Fonseca et al. have proposed a new model for anamnestic immune responses, an ‘outside-in’ immune response [18]. Historically, the memory T cell population was divided into T_CM_ and T_EM_. T_CM_ reside predominantly in SLOs and differentiate into various T cell subtypes upon reactivation with antigen. T_EM_ patrol in blood and T_RM_ surveil in NLT [65]. The ‘inside-out’ model (left panel, Figure 2) describes primary immune responses. T_RM_ were proposed to reside permanently in NLT, where they would proliferate locally upon antigen encounter and thus provide protection [21,22]. This model can be seen as ‘inside-out’, because T_CM_ are activated in SLOs, i.e., ‘deeper’ tissues, after which T_CM_ and T_EM_ would shape the systemic recall immune response, proliferating and migrating towards affected tissue sites. The proposed ‘outside-in’ model (right panel, Figure 2) suggests that T_RM_ (which are activated in barrier tissues) participate in the systemic immune response by potentially contributing to the circulating memory T cell pool, egressing and differentiating into T_CM_ and T_EM_ (epigenetic plasticity), and even populating other tissues [18,66]. Moreover, there has been further speculation that this systemic distribution of T_RM_ might contribute to a broad protection against pathogens, if pathogens were to escape local defense [66]. Circulating ex-T_RM_ populations could also serve to re-populate NLT in the event of depletion of the local T_RM_ population [18].

### 3.3. Indications for Recirculation of T_RM_ in Chronic Inflammatory Disease

As previously mentioned, studies have highlighted the presence and role of T_RM_ in several tissues permeated with chronic inflammation. The outside-in model described above, proposes that circulating ex-T_RM_ could be contributing to the expansion of inflammatory sites in affected tissue, as has been suggested in the several disease settings such as psoriasis [41,53,54], IBD [36], and JIA [35,36]. Flow cytometry was implemented to examine the phenotype of CD4^+^ T cells in the synovium of patients with active JIA. These cells displayed distinct characteristics compared to T cells in circulation, including increased expression of CD69, PD1, CTLA-4, Ki-67, CCR5, and CCR6, and decreased expression of CCR7 [35,36,67]. It was hypothesized that if a small fraction of the pro-inflammatory synovial CD4^+^ T cells would leave the inflamed synovium, a population of blood T cells with similarities to synovial T cells should exist. Indeed, a small subset of PB T cells that displayed the same characteristics as synovial T cells was identified, albeit with lower CD69 expression. These T cells were labelled ‘circulating pathogenic lymphocytes’ (CPLs). To further investigate these CPLs, next-generation sequencing of the TCRβ chain was performed. Both synovial T cells and CPLs displayed decreased clonal diversity compared to total blood CD4^+^ T cells. Moreover, clustering of TCRβ sequences showed that CPLs shared more clonotypes with their synovial counterparts than other blood T cells. It was also demonstrated that CPLs had increased capacity for the production of IL-17, IFN-γ, and TNF-α compared to other non-CPL CD4^+^ T cells in the blood. This suggests that these cells retain their active, pro-inflammatory profile in circulation. In addition, CPLs correlated with disease activity in JIA and RA and also with resistance to therapy in JIA (Figure 3) [67].

Tregs are a crucial part of suppressive immunity and have been shown to differ phenotypically and functionally in non-diseased and diseased states [60,68,69,70]. It has been shown that Tregs are part of the T_RM_ compartment i.e., CD69-expressing Treg in synovial, uterus, and gut tissue [60,71]. Tregs from chronic inflammatory settings are phenotypically/functionally different from healthy tissue Treg [36,60,70,72] and it has also been postulated that Tregs can re-circulate [72]. By studying the PB of JIA patients, it was observed that a specific subset of Tregs, which was positive for HLA-DR, was significantly increased in patients who did not respond to therapy compared with patients who reached inactive disease in response to therapy. These Tregs were as labelled ‘inflammation associated Tregs’ (iaTreg). iaTregs showed increased CCR5, Ki-67, and CTLA-4, compared to other Tregs, indicating that they can home to tissue and have been recently activated. Moreover, next-generation sequencing of the TCRβ chain demonstrated that iaTregs were clonally related to synovial Tregs, more than to Tregs in the blood. Finally, expansion of iaTregs was observed in the PB of adults with active RA. Together, these data show that a specific subset of Tregs that is clonally more similar to synovial Tregs compared to other blood Treg, is expanded in the PB of both JIA and RA patients (Figure 3) [72]. When examining T_RM_ in conjunction with Treg populations, though CD69 and CD103 are adequate to construe it, there is some selectivity within different compartments in a tissue (i.e., increased CD103 expression in the epithelial compartment compared to the lamina propria in the gut). Tissue migration capacities by upregulation of integrins is induced in the lymph nodes, and can be driven by inflammatory stimuli (such as infections) but also less harmful antigens derived from food or commensals (as seen in early life seeding of naïve peripheral tissues such as the gut). Once in the tissue, cells may differentiate/adapt through tissue-specific, antigen, and inflammatory cues both for Treg and non-Treg cells [60,69,70,71,73]. It is currently unknown how selective maintenance in different tissues is regulated.

Other T cell populations in SF and PB of JIA patients were also investigated. Several clusters of CD4^+^ T cells in the SF were identified, two of which expressed T_RM_ associated genes. These T_RM_-like cell clusters were enriched in the SF of JIA patients compared to healthy controls. One of the two clusters did not only display gene expression associated with T_RM_, but also with circulating T cells. This subset expressed IL-10, Eomes, PRF1, TNFRSF9, and genes encoding granzymes, which indicate resemblance to type 1 regulatory T (Tr1)-like cells. Furthermore, analysis of the PB of JIA patients led to the identification of a small blood CD4^+^ T cell cluster that mirrored gene expression of the Tr1-like cell resembling cluster in the SF and was clonally related to the Tr1-like cluster [36].

Taken together, findings indicate that there is recirculation of both Treg and pathogenic CD4^+^ T cells between blood and SF in JIA patients. Recirculation may contribute to the evolution of the disease over time, i.e., the involvement of multiple joints over the disease course. It may even be responsible for spreading of the disease to different organs, such as the gut (or the skin). Qaiyum et al. identified a population of CD103^+^β7^+^CD49a^+^CD29^+^CD8^+^ T cells that was enriched in the SF of ankylosing spondylitis patients compared to PB. Because the β7 integrin is a gut homing molecule, and ankylosing spondylitis is associated with IBD, the authors speculated that the population of T cells they identified might originate from the gut [38]. This notion was further investigated by Guggino et al., where paired samples from gut, synovial tissue, SF and PB from ankylosing spondylitis patients and controls were examined. Using flow cytometry, they observed that CD8^+^CD69^+^CD103^+^ T_RM_ were expanded in not only SF, but also in gut and PB of ankylosing spondylitis patients compared to healthy controls. The majority of these cells in SF and PB expressed β7 integrin. These data support the hypothesis that T_RM_ from the gut recirculate to PB and inflamed joints in patients with ankylosing spondylitis (Figure 3) [74].

As such, disease-associated T_RM_ entering other tissues via circulation might contribute to off-site or systemic disease-associated symptoms that often occur in autoimmune diseases. However, current data only provide indications for this, and further elucidation of these circulating pathogenic T_RM_ populations and their association with disease symptoms is warranted.

## 4. Summary and Outlook

Post-pathogen exposure, T_RM_ become established in peripheral tissue. In the case of re-exposure to antigen T_RM_, as is well known, mount an immune response at the site of infection, or as we suggest, in settings of chronic inflammation, are capable of not only taking on an aggressive effector function but egressing and through developmental plasticity, contributing to the effector memory and effector cell pools. In this instance, they may be comparable to TIL-like cells in cancer settings, functionally overlapping with T_EM_. Interestingly overlap in phenotype and transcriptome has been demonstrated for inflamed joint- and tumor-derived Treg [69]. The transient expression of CD69 upon activation may be a gradient along which we can discern T_RM_ from T_EM_, though TCR sequencing and single cell transcriptomics may provide a robust additional layer of characterization. Though the exact trigger of ex-T_RM_ in autoimmunity remains yet unknown, further examination of the ex-T_RM_/egressed T_RM_-like subsets may provide insight into the mechanism(s) by and the motive for which these cells have entered circulation.

Novel concepts of T_RM_ recirculation and pro-inflammatory, effector-like functions of T_RM_ may add to the paradigm of diagnosing, treating, and understanding the etiology of autoimmune diseases and chronic inflammation. T_RM_ cells are crucial in maintaining homeostasis, providing a stringent immune front in tissues. However, their involvement in chronic inflammation and the degree of damage induced by relapsing episodes abrogate efforts made to manage diseases successfully. The heterogeneity of the T_RM_ population alludes to the difficulty of aptly identifying and potentially therapeutically targeting these mostly tissue embedded cells. Moreover, the enrichment of T_RM_ found in non-lesional tissue of patients suggests that T_RM_ may contribute to predisposition for chronic inflammation, resulting in a higher likelihood of relapsing episodes that require further medical and lifestyle interventions. The burden of disease in conditions with a relapsing-remitting course is understandably difficult for patients and also challenging for clinicians to manage and/or treat. The concept of T_RM_ recirculation further denotes the contribution of these cells in spreading of the disease and may also add to off-site or systemic disease-associated symptoms that often occur in autoimmune diseases. For instance, in the case of JIA-associated uveitis, further investigation into the critical role of immune cells and response in this setting would vastly benefit patients from debilitating long-term ocular damage [75].

As such, early targeted therapy could be an interesting approach to halt disease progression. Targeting T_RM_ would not be a clear-cut approach since there are several variables that prove difficult to estimate in terms of their identification, placement, function, and numbers. For example, what would be an ideal number of T_RM_ to target in autoimmune diseases? Additionally, the potential side effects of targeting T_RM_ are unknown and may be detrimental as these cells are immune sentinels and vital for homeostasis. Targeting T cells in tissue may also prove to be technically difficult, but this may be circumvented by targeting/blocking (re)migration of these immune cells. In CD, the biological therapeutic Vedolizumab effectively targets immune cells via α4β7 (a marker for homing to gut tissue). The standard of care for patients with CD is a treat-to-target approach, with an early, aggressive attempt at inflammatory control. However, the majority of patients are intolerant to conventional therapy and gradually intolerant to TNF-α therapy [76]. In the GEMINI-LTS trial, patients treated with Vedolizumab at four-week intervals resulted in 83% [*n* = 100/120] and 89% [*n* = 62/70] of patients in remission after 104 and 152 weeks, respectively, showing remarkable improvement and positive long-term response [77]. Similarly, MS patients can be effectively treated with the α4-integrin inhibitor Natalizumab, blocking T cell migration over the blood-brain barrier, as well as to the intestines. However, this treatment is also linked to progressive multifocal leukoencephalopathy (an opportunistic viral infection), stressing the delicate balance in such settings [78].

Identification of disease-exacerbating T_RM_ cells in circulation will not only open avenues for novel therapeutics but may also allow crucial monitoring (biomarkers) of disease progression. For instance, the disease course of JIA is unpredictable. Currently, JIA patients are only ‘allowed’ to be treated with biologicals (e.g., TNFα blockers or IL-6 blockade) when they are unresponsive to the first-line treatment methotrexate. However, unresponsiveness to methotrexate occurs in 30–50% of patients. If resistance to methotrexate therapy could be reliably predicted, patients could be treated with biologicals instantly. The window of opportunity, a period shortly after disease onset during which an optimal effect of treatment can be achieved, would then be utilized. Recirculating ex-T_RM_ may serve as prognostic biomarkers of disease severity in chronic inflammation and autoimmune disease, and may even predict resistance to therapy.

## Figures and Tables

**Figure 1 cells-10-00409-f001:**
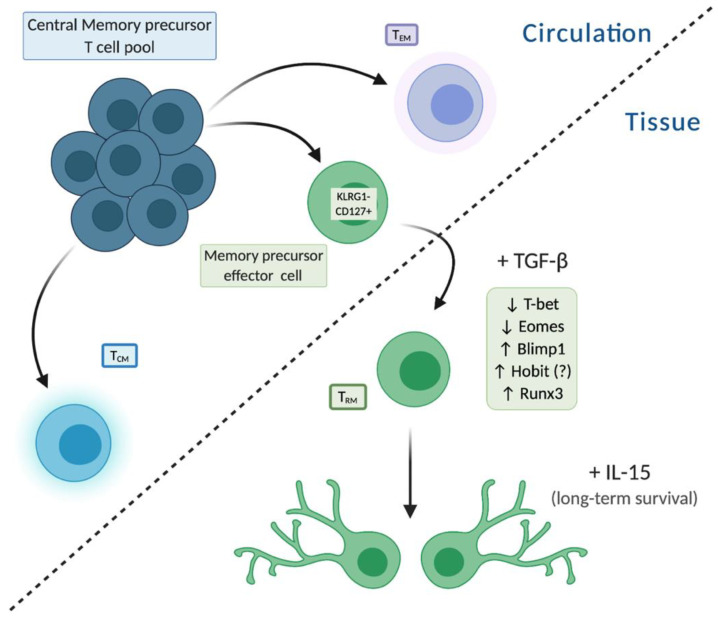
Key cytokines and transcription factors for the development of tissue–resident memory T cells (T_RM_). T_RM_ are derived from memory precursors effector cells that express low KLRG1 and high CD127. Analysis of development has shown that T_RM_ displayed plasticity intermediate between tissue central-memory (T_CM_) and tissue effector-memory (T_EM_), and are not as terminally differentiated as previously believed. The central cytokine that induces T_RM_ differentiation is transforming growth factor β (TGFβ). TGFβ induces downregulation of T-bet and Eomes, which is essential for T_RM_ formation. Blimp1, Hobit, and Runx3 play a role in T_RM_ differentiation as well. Upregulation of Hobit is observed in mice T_RM_, but only in a subset of human T_RM_.

**Figure 2 cells-10-00409-f002:**
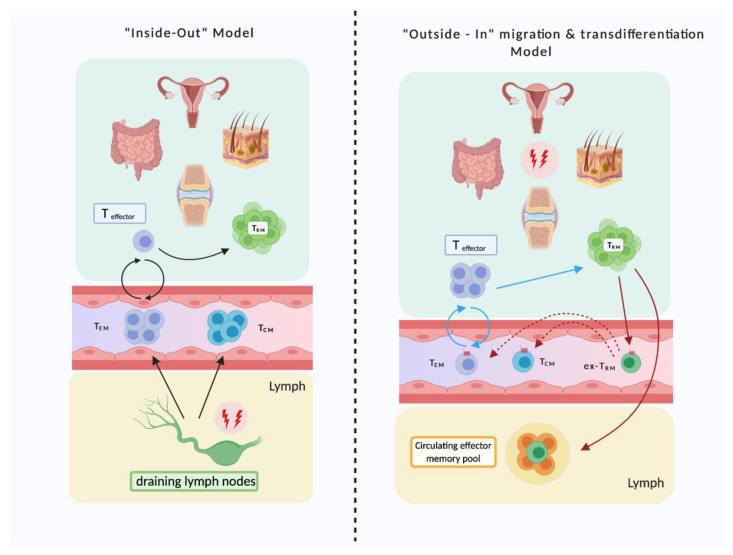
Models of the recall immune response by memory T cells. The left panel shows the ‘inside-out’ model, where T_RM_ are local tissue mediators of the anamnestic immune response, proliferating inside their tissue upon reactivation. T_CM_ and T_EM_ constitute the systemic recall response in this model. T_CM_ are activated in deeper tissues i.e., draining lymph nodes at site of infection, proliferating and migrating towards affected tissue sites. The right panel shows the recently proposed ‘outside-in’ model. In this mode anamnestic immune response is generated at site of infection. T_RM_ are activated at barrier tissues and migrate inwards, to lymph and circulation, with the potential to differentiate into T_CM_ and T_EM_ (highlighted by the arrows in red). The blue arrows represent the T cell traffic under chronic inflammatory (permanently activated) states [18,21,22,65,66].

**Figure 3 cells-10-00409-f003:**
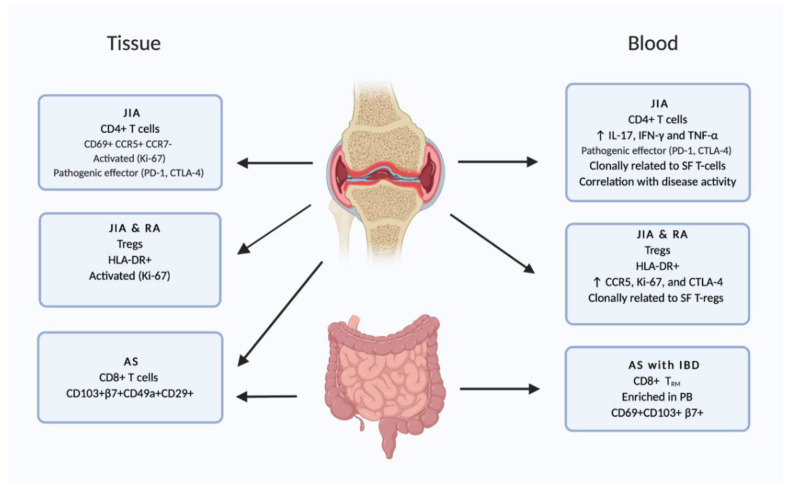
T cells resembling tissue–resident memory T cells (T_RM_) from various tissues have been identified. Subsets of memory T cells with characteristics of T_RM_ were identified in the circulation of Juvenile Idiopathic Arthritis (JIA), RA, and AS (with inflammatory bowel disease (IBD)) patients. These cells displayed an activated, effector pathogenic phenotype, and were clonally related to their tissue counterparts. This suggests that these cells may be ex-T_RM_, having egressed from the tissue of origin. Cells resembling synovial fluid (SF) Tregs were also identified in circulation.

**Table 1 cells-10-00409-t001:** Summary of the characteristic transcriptional profile of tissue–resident memory T cells (T_RM_). In this table, a selection of transcripts that are part of the key transcriptional profile displayed by T_RM_ is shown. It must be noted that the markers displayed in this table do not occur on every population of T_RM_.

Transcript	Gene	Up- (↑) or Down (↓) Regulated	Expression-Human vs. Mouse	References
	Surface markers			
**CD69**	*Cd69*	↑	Both	[3,9,10]
**CD103**	*Itgae*	↑	Both	[3,9,10]
**CD49a**	*Itga1*	↑	Both	[3,9,10]
**CD101**	*Cd101*	↑	Both	[3]
**CD62L**	*Sell*	↓	Human	[3,9,10]
**CXCR6**	*Cxcr6*	↑	Both	[3,10]
**CX3CR1**	*Cx3cr1*	↑	Human	[3]
**CCR7**	*Ccr7*	↓	Human	[3,9]
**PD-1**	*Pdcd1*	↑	Human	[3,10]
**S1PR1**	*S1pr1*	↓	Both	[3,9,10]
	Intracellular proteins			
**DUSP6**	*Dusp6*	↑	Both	[3,10]
**KLF2**	*Klf2*	↓	Both	[3,10]
**KLF3**	*Kl3*	↓	Human	[3,10]
**Eomes**	*Eomes*	↓	Human	[9,11]
**T-bet**	*Tbx21*	↓ *	Human	[11]
**Blimp1**	*Prdm1*	↑	Mouse	[12]
**Hobit**	*Zfp683*	↑	Mouse	[12]
**Runx3**	*Runx3*	↑	Mouse & Human TIL **	[13]
**Id3**	*Id3*	↑	Mouse	[14]
**Nr4a1**	*Nr4a1*	↑	Human	[9]

* T-bet is downregulated, but some expression of T-bet is required for T_RM_ survival [9]. ** TIL-tumor infiltrating lymphocytes.

**Table 2 cells-10-00409-t002:** Populations of tissue–resident memory T cells (T_RM_) that are enriched in barrier and non-barrier autoimmune diseases. This table provides an overview of the populations of T_RM_ that were observed to be significantly enriched in several inflammatory diseases. The fourth column states the markers that were used to distinguish the enriched T_RM_ population in that particular study.

Chronic Inflammatory Diseases.	Tissue Enrichment	Specific Markers/Factors/Genes Associated	Characteristics of Enriched T_RM_ Population	Ref
**Psoriasis**	Skin	IL-17	CD49a^+^ CD103^+^ CD8^+^	[40]
			CD103^+^ CD8^+^	[41]
**Vitiligo**	Skin	IFN-γ, CXCR3	CCR7^−^ CD69^+^ CD103^+^ CD8^+^	[42]
			CD69^+^ CD103^+^ CD8^+^	[43]
**Atopic dermatitis**	Skin	CCL1, IL13, IL26, ANXA1, ANXA2	CD69^+^ CD103^+^ CD8^+^	[44]
**Alopecia areata**	Skin	ITGAE	CD103^+^ CD8^+^	[45]
			CD69^+^ or CD103^+^ (CD4 or CD8 not specified)	[46]
**Inflammatory bowel disease**	Gut	CD161, β7	CCR7^−^ CD69^+^ CD4^+^	[47]
		CXCR6, CD101, KLF2lo	CD69^+^ CD103^+^ (CD4 or CD8 not specified)	[24]
**Type 1 diabetes**	Pancreas	IFN-γ, IL-18, IL-22	CD69^+^ CD103^+^ CD8^+^	[30]
**Sjögren’s syndrome**	Nerve/Connective tissue	IFN-γ	CD69^+^ CD103^+^ CD8^+^	[31]
**Lupus nephritis**	Connective tissue	JAK/STAT, TNF-α, IFN-γ	CD103^+^ CD8^+^	[32]
**Multiple sclerosis**	Central nervous system	CD69^−^	CD103^+^ CD8^+^	[34]
		CXCR6, Ki67	PD-1^+^ CD44^+^ CD49a^+^ CD69^+^ CD103^+/–^ CD8^+^	[33]
**Juvenile idiopathic arthritis**	Joint	ITGAE, ITGA, CXCR6	PD-1^+^ CD69^+^ CD8^+^	[35]
		DUSP6, ITGAE, CXCR6, PD-1	PD-1^+^ CD8^+^, PD-1^+^ CD39^+^/CD161^+^ CD4^+^ and PD-1^+^ CD39^−^/CD161^−^ CD4^+^	[36]
**Psoriatic arthritis**	Joint/Skin	PD-1, ITGAE, ZNF683, CXCR6, β7, CLA, CD49a	IL-17A^+^ CD69^+^ and/or CD103^+^ CD8^+^	[37]
**Ankylosing spondylitis**	Connective tissue/Joint	β7, CD29, IL-10, CXCR6	CD49a^+^ CD103^+^ CD8^+^ T cells	[38]
**Rheumatoid arthritis**	Connective tissue/Joint	PD-1, Blimp, 1, CD44	PD-1^+^ CXCR5^−^ CD69^+^ CD4^+^	[48]
		CXCR6, CD49a, CD101, PD-1, Ki-67	CD69^+^ CD103^+/−^ CD8^+^	[39]

## Data Availability

No new data were created or analyzed in this study. Data sharing is not applicable to this article.

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
