# Peer review of "Tissue–Resident Memory T Cells in Chronic Inflammation—Local Cells with Systemic Effects?"

_cells, 2021, doi:10.3390/cells10020409_

Round 1

Reviewer 1 Report

This manuscript documents the comprehensive review of tissue resident memory T cells, especially focusing on their development and potential functions in autoimmune diseases. Furthermore, this review discusses the role of circulating “ex-TRM” in autoimmune diseases. So far, it remains unclear whether TRM is involved in the pathogenesis of autoimmune diseases. In general, TRM are established in peripheral tissues following the resolution of pathogen infection. In addition, TRM barely require the residual antigens for the long-term lodgment in peripheral tissues. Therefore, it would be conceivable that TRM-like cells in autoimmune diseases are likely to be resemble to TIL-like cells in cancer regions. In this regard, the difference between TEM and TRM also remains obscure. In terms of “ex-TRM”, the direct evidence of “ex-TRM” in autoimmune diseases is currently missing. So far, some virus re-infections are known to trigger the release of TRM from peripheral tissues into blood circulation. However, it entirely remains unknown the trigger of “ex-TRM” in autoimmune diseases. Please discuss the mentioned-above points.

Reviewer 2 Report

Samat and colleagues provide a timely review of tissue-resident memory T (Trm) cells with a special focus on the literature dealing with Trm cells in chronic inflammatory diseases. Since their discovery, the field of Trm has developed into a highly competitive are in immunology with relevance to tissue health, infection control, immunological diseases and cancer. The present review focuses on the most recent advances and links progress in Trm research with chronic inflammation. Such a tailored review has not been published recently to my knowledge.

The review cites many key papers, most of them of recent date, and provides a balanced account on past research. It is well written and provides an interesting perspective for the non-specialist reader with an interest in adaptive immune memory. Below are a couple of suggestions that may further benefit the understanding of this review article.

  1. If at all possible, please modify Fig 2 to include a panel depicting the situation of T cell traffic under chronic inflammatory (i.e. permanently activated) conditions (after all this is the major topic of this article)
  2. Legend to Fig 2 should state clearly where is has been taken from (group of Masopust)
  3. Fig 3 needs clarification because of the inclusion of Treg cells. Obviously, when talking about autoimmunity or chronic inflammation there is no way around Treg cells. But the distinction of Treg cells and disease-associated Tconv cells is not discussed at all. Since chronic inflammation has to do with lack of immune control, it would be useful for the reader to understand how Treg cells fit in the current concept of Trm cells. In fact, are they indeed part of the Trm compartment? Do they also recirculate? Are they missing in chronic inflammation? Are they functionally/phenotypically different from Treg cells in healthy tissue?
  4. On a different note, perhaps the above discussion could include the authors’ view about how tissue selectivity of Trm cells is generated (vis-à-vis local Treg cells) and, more importantly, maintained. Obviously, CD69 and CD103 are inadequate to explain tissue selectivity. I understand that this topic is still in need of clarification.
  5. In my view, Chapter 3.3 is the most interesting part, which I would like to know more about. For instance, could Fig 3 be modified to include a list of the distinct features characterizing Trm cells in individual inflamed tissues (digestive tract, lungs, skin, joints) as opposed to healthy tissues. Especially human skin is missing, a tissue whose Trm compartment we know much about. Of course, by focusing on the human situation, I understand that the relevant literature is limited.
  6. Chapter 3.1 about Trm recirculation is too lengthy, especially considering that is based mainly on 3 papers from the same group. I suggest to cut this section by leaving out the experimental details.
  7. I would have liked to hear the authors’ view about the initial generation of Trm cells that populate “naïve” peripheral tissues in the first place. For instance, what is the evidence that Trm cells are derived from memory precursor cells as opposed to effector cells recruited to tissue sites in response to local inflammation (Fig 1)? And what are the tissue factors driving Trm cell generation?
  8. How do the authors differentiate between activated (CD69+CD62L-) effector T cells found in engaged LNs and CD69+ Trm cells? This question is relevant for the appropriate choice of phenotypic markers when describing Trm cells in original tissues and their descendants in blood.
